# Stochastic Arbitrage Opportunities: Set Estimation and Statistical Testing

**Stelios Arvanitis [1],* and Thierry Post [2]**

1. Department of Economics, Athens University of Economics and Business, 104 34 Athens, Greece
2. Graduate School of Business, Nazarbayev University, Astana 010000, Kazakhstan; thierrypost@hotmail.com
* Correspondence: stelios@aueb.gr

**Abstract:** We provide a formal statistical theory of consistent estimation of the set of all arbitrage portfolios that meet the description of being a stochastic arbitrage opportunity. Two empirical likelihood ratio tests are developed: one for the null that a given arbitrage portfolio is qualified, and another for the alternative that the portfolio is not qualified. Apart from considering generalized concepts and hypotheses based on multiple host portfolios, the statistical assumption framework is also more general than in earlier studies that focused on special cases with a single benchmark portfolio. Various extensions and generalizations of the theory are discussed. A Monte Carlo simulation study shows promising statistical size and power properties for testing the null, for representative data dimensions. The results of an empirical application illustrate the importance of selecting a proper blocking structure and moment estimation method.

**Keywords:** portfolio analysis; arbitrage portfolios; asset pricing; asymptotic statistics; empirical likelihood

## 1. Introduction

Stochastic dominance (SD) is a celebrated investment decision-making criterion that avoids specification error for investor risk preferences by relying on a general class of utility functions instead of a parametric specification. It is particularly appealing when the Gaussian distribution gives a poor approximation, for example, for analyzing dynamic strategies and derivative securities.

The historical evolution of the theory, methodology and application of SD is described in detail in the monographs by [1–3].

The application of SD was traditionally focused on the pairwise comparison of two alternatives (e.g., two securities, two portfolios, or two funds), but over time, concepts and methods were also developed for the multivariate analysis of a multitude or a continuum of alternatives for asset pricing and portfolio analysis.

A milestone on this path was the modeling of non-smooth restrictions on the lower partial moments of a portfolio relative to a given benchmark, which facilitated the development of various problem formulations and optimization algorithms for building a portfolio that stochastically dominates a given benchmark [4–8]. Ref. [9] develop a statistical theory based on the empirical likelihood (EL) method for the associated efficient set and the optimal portfolio.

The concepts and methods for multivariate analysis have made it possible to apply SD to realistic, large-scale investment problems. For example, ref. [10] apply it to test pricing efficiency in the market for stock index options; ref. [11] apply it to tactical asset allocation across equity industries; and ref. [12] apply it to construct active combinations of stock index options.

Ref. [13] further generalize the concepts and methods by accounting for uncertainty about the relevant benchmark portfolio or host portfolio. To account for this type of uncertainty, they introduce the concept of a general stochastic arbitrage opportunity (SAO): a zero-cost investment portfolio (arbitrage portfolio, hedge portfolio or overlay) that enhances all hosts that can be constructed from a set of base assets (instead of a single benchmark portfolio) for all elements of a general set of utility functions.

The SAO concept generalizes the classical concept of a pure arbitrage opportunity. Since the latter generally suffers from a lack of discriminatory power in incomplete markets, generalized arbitrage concepts have been developed, such as [14–17]. The distinguishing feature of the SAO concept is that it builds on the generally accepted SD concept, and it does so without assuming that the host is unique and known.

The generalized concepts and methods have several potential applications in asset pricing and portfolio management. Examples of relevant applications include the following:

(a)   In empirical asset pricing tests of market portfolio efficiency, the weights of the market portfolio are generally latent, introducing the risk of specification error. The SAO framework allows for a robust rejection of efficiency if it can be established that there exist overlays that enhance a set of multiple market portfolio proxies.

(b)   In the performance evaluation of mutual funds and hedge funds, the relevant style benchmark often cannot be estimated with high accuracy due to the use of short time series and changes in the investment style over time. The proposed tests can identify robust outperformance if it can be established that a given fund enhances a set of multiple style benchmarks.

(c)   In asset allocation across specialized funds for multiple asset classes, the dominance of individual funds over their single-asset benchmarks unfortunately does not guarantee improvement for multi-asset investors. A potential remedy is to restrict attention to allocations that stochastically enhance all relevant mixtures of multiple single-asset benchmarks.

For empirical estimation and statistical inference, ref. [13] sketch a theory of consistent estimation of the set of all overlays that qualify as an SAO and consistent testing of whether or not a given overlay is qualified. The present study provides a formalization of that statistical theory, additional discussion about extensions and generalizations, and a forward-looking research agenda.

We demonstrate how the analysis by [9] of optimization under SD restrictions can be generalized to arbitrage portfolio choice under stochastic enhancement restrictions for a general specification of the host portfolio set (K) and the utility class ($\mathcal{U}$). We utilize a generalization of the weak independence assumption of [9] in order to obtain versions of set consistency for the empirical SAOs, notably avoiding the use of (arbitrary) tolerance parameters. The derivations are novel; they are, among other things, based on the construction of a set of approximate solutions to the empirical optimization problem that satisfy restricted versions of the empirical dominance inequalities.

We also show how to perform statistical inference about the classification of an overlay as an SAO/not an SAO using blockwise empirical likelihood ratio (BELR) tests, in the spirit of the tests for the simpler concepts of dominance/non-dominance, efficiency/inefficiency and optimality/non-optimality by [18–20]. Two empirical likelihood ratio tests are developed: one for the null that a given overlay is a true SAO, and another for the alternative that it is not a true SAO. We furthermore introduce new robust versions of the testing procedures that allow for the possibility that the set of binding dominance inequalities is infinite.

Although the focus of this study is on the general case with a set of hosts and a set of overlays, the results naturally also apply to simpler cases in which one of the two sets is a singleton. The simplest special case arises for the pairwise analysis between two given standard portfolios or, equivalently, the analysis of one overlay and one host. Other special cases are the efficiency analysis of a given standard portfolio and portfolio optimization with SD constraints relative to a given benchmark portfolio.

The remainder of this study is organized as follows. Section 2 summarizes the relevant economic concepts. Section 3 develops the statistical theory. Section 4 presents the results of a Monte Carlo simulation experiment. Section 5 summarizes the results of an empirical exercise that complements the empirical analysis by [13] of equity factor portfolios and equity industry portfolios. Section 6 concludes by discussing various extensions and generalizations of the theory, and ideas for further research.

## 2. Stochastic Arbitrage Opportunities

### 2.1. Host Portfolios, Arbitrage Portfolios and Combined Portfolios

The analysis is based on a standard one-period investment problem with $N$ elementary assets with risky payoffs at a future investment horizon, $\boldsymbol{x} := (x_1 \cdots x_N)^{\mathrm{T}} \in \mathbb{R}^N$, $N < \infty$. The prices of the assets are externally determined and collected in the price vector $\boldsymbol{p} \in \mathbb{R}^N$. Payoffs may be transformed to gross investment returns, $\boldsymbol{r} := \mathrm{diag}^{-1}(\boldsymbol{p})\boldsymbol{x}$, without loss of generality.

The joint probability distribution of the payoffs is represented by $\mathbb{P}$; its support is a subset of $\mathcal{X}^N$, where $\mathcal{X} := [a, b]$, $-\infty < a < b < \infty$. The distribution may be conditional on the conditions of the general economy and the financial markets. Unless an empirical version of $\mathbb{P}$ is specified, expectations and probabilities are evaluated with respect to $\mathbb{P}$.

Utility functions $u : \mathbb{R} \to \mathbb{R}$ are used to evaluate alternatives based on the expected utility of the payoffs. To avoid specification errors, we eschew the choice of a particular functional form. Instead, a set of functions, $\mathcal{U}$, is defined using general functional properties. The set is assumed to be uniformly bounded and convex. It is assumed that utility functions are continuously differentiable, strictly increasing and strictly concave. Without loss of generality, the functions are standardized such that $u(b) = 0$ and $u'(a) = 1$.

All utility functions that fit the above description can be represented using strictly positive combinations of elementary [21] functions $v_{2;\phi}(x) := -(\phi - x)_+$, $\phi \in \mathcal{X}$. The totality of these combinations form the class of utilities $\mathcal{U}_2 := \{u(x) = \int_{\mathcal{X}} v_{2;\phi}(x) dW(\phi); W \in \mathcal{W}\}$. Here, $\mathcal{W}$ is the set of increasing cumulative weighting functions $W : \mathcal{X} \to [0, 1]$.

$\mathcal{U}$ is endowed with the sup metric to facilitate the asymptotic theory of the inferential procedures in Section 3. The asset pricing results in ([13], Theorem 3.3.1) rely on first-order optimality conditions that require a stronger topology, like $W^{1,\infty}(\mathbb{R})$; see [22].

The utility set $\mathcal{U}$ is not necessarily closed because singular utilities obtained as limits of strictly increasing and concave functions are inconsequential for the analysis. The closure of the set of utilities, $\mathrm{cl}(\mathcal{U})$, is however useful for the numerical analysis and the analysis of the limiting behavior of empirical constructs.

Two distinct and complementary portfolio sets are used to describe the available portfolio possibilities: a set of benchmarks or hosts, denoted by $\mathrm{K} \subseteq \mathbb{R}^N$, and a set of overlays, denoted by $\Delta \subseteq \mathbb{R}^N$. A description of these two portfolios sets follows.

Host portfolios are standard portfolios that require a strictly positive net investment. The asset holdings vector $\boldsymbol{\kappa} \in \mathrm{K}$ should obey the budget condition $\boldsymbol{p}^{\mathrm{T}}\boldsymbol{\kappa} = c$ for budget $c > 0$. Asset holdings may be transformed to portfolio weights $\boldsymbol{\lambda} := c^{-1}(\boldsymbol{p} \odot \boldsymbol{\kappa})$.

It is assumed that $\mathrm{K}$ is closed and bounded, but convexity is not required. In case of non-convexity, the convex hull is denoted by $\mathrm{conv}(\mathrm{K}^{(0)})$; in case of convexity, the set of extreme elements is denoted by $\mathrm{K}^{(0)}$. If $\mathrm{K}$ takes a polyhedral shape, then the number of extreme elements is finite; in this case, the continuous set $\mathrm{K} = \mathrm{conv}(\mathrm{K}^{(0)})$ can be replaced by $\mathrm{K}^{(0)}$ for numerical purposes.

In contrast to standard portfolios, arbitrage portfolios $\boldsymbol{\delta} \in \Delta$ are mixtures of short positions and long positions that are self-financing, or $\boldsymbol{p}^{\mathrm{T}}\boldsymbol{\delta} = 0$. If an overlay is added as an overlay to a host $\boldsymbol{\kappa} \in \mathrm{K}$, then the combined portfolio $\boldsymbol{\lambda} = (\boldsymbol{\kappa} + \boldsymbol{\delta})$ naturally is a standard portfolio ($\boldsymbol{p}^{\mathrm{T}}\boldsymbol{\lambda} = c$). The vector subspace sum $\Lambda_0 := \mathrm{K} + \Delta$ defines the totality of all feasible combined portfolios. Without loss of generality, the asset holdings may equivalently be formulated using weights in the combined portfolio, or $\boldsymbol{\gamma} := c^{-1}(\boldsymbol{p} \odot \boldsymbol{\delta})$.

The arbitrage portfolio set is assumed to be a bounded and convex polytope that is defined by $R$ linear inequality restrictions: $\Delta := \{\boldsymbol{\delta} \in \mathbb{R}^N : \mathbf{A}\boldsymbol{\delta} \leq \boldsymbol{a}\}$; here, $\mathbf{A}$ is a $(R \times N)$

coefficient matrix, and $a$ is a $(R \times 1)$ column vector. It is assumed that the 'passive' solution $0_N \in \Delta$ is the default choice if SAOs do not exist.

Although the focus here is on general portfolio sets, the analysis also applies to simpler cases in which K and/or $\Delta$ is singleton. To make a pairwise comparison of two given standard portfolios $\kappa_1$ and $\kappa_2$, we may simply apply our framework to K = $\{\kappa_1\}$ and $\Delta = \{\kappa_2 - \kappa_1\}$; to analyze the efficiency of a given standard portfolio $\kappa_1$ relative to standard portfolio set $\Lambda$ and find an alternative portfolio $\lambda \in \Lambda$ that stochastically dominates $\kappa_1$, we may simply set K = $\{\kappa_1\}$ and $\Delta = \Lambda - \kappa_1$.

*2.2. Stochastic Arbitrage Opportunities*

Ref. [13] introduce two versions of the concepts of stochastic enhancement and SAO: a strict version for asset pricing theory and a weak version for numerical purposes.

For brevity, the increase in expected utility that is achieved by laying an arbitrage portfolio over a standard portfolio will be represented in this study by $D(u, \kappa, \delta, \mathbb{P}) := \mathbb{E}[u(x^{\mathrm{T}}(\kappa + \delta))] - \mathbb{E}[u(x^{\mathrm{T}}\kappa)]$ for $(u, \kappa, \delta) \in \mathcal{U} \times K \times \Delta$.

**Definition 1.** *(Strict SAO). An arbitrage portfolio $\delta \in \Delta$ (strictly) stochastically enhances a given benchmark $\kappa \in$ K, or $(\kappa + \delta) \succ_{(\mathcal{U}, \mathbb{P})} \kappa$, if $D(u, \kappa, \delta, \mathbb{P}) > 0$ for all $u \in \mathcal{U}$. It is a (strict) SAO if such enhancement is achieved for all $\kappa \in$ K. The set of all feasible (strict) SAOs is given by*

$$\Delta_{(\mathcal{U}, \mathrm{K}, \mathbb{P})}^{\mathrm{SAO}} := \{\delta \in \Delta : D(u, \kappa, \delta, \mathbb{P}) > 0 \; \forall (u, \kappa) \in \mathcal{U} \times \mathrm{K}\}. \quad (1)$$

**Definition 2.** *(Weak SAO). An arbitrage portfolio $\delta \in \Delta$ (weakly) stochastically enhances a given benchmark $\kappa \in$ K, or $(\kappa + \delta) \succsim_{(\mathcal{U}, \mathbb{P})} \kappa$, if $D(u, \kappa, \delta, \mathbb{P}) \geq 0$ for all $u \in \mathrm{cl}(\mathcal{U})$. It is a (weak) SAO if such enhancement is achieved for all $\kappa \in$ K. The set of all feasible (weak) SAOs is given by*

$$\Delta_{(\mathcal{U}, \mathrm{K}, \mathbb{P})}^{\mathrm{WAO}} := \{\delta \in \Delta : D(u, \kappa, \delta, \mathbb{P}) \geq 0 \; \forall (u, \kappa) \in \mathrm{cl}(\mathcal{U}) \times \mathrm{K}\}. \quad (2)$$

*Thus, SAOs are solutions to the below semi-infinite system of inequalities:*

$$D(u, \kappa, \delta, \mathbb{P}) \geq 0 \; \forall (u, \kappa) \in \mathrm{cl}(\mathcal{U}) \times \mathrm{K}; \quad (3)$$
$$\delta \in \Delta.$$

To perform numerical analysis using finite mathematical programming problems, it is useful to employ finite discretizations of $\mathbb{P}$, $\mathrm{cl}(\mathcal{U})$ and K. Various common and general discretizations are discussed below.

Monte Carlo simulation methods and lattice models can be used to discretize continuous distributions $\mathbb{P}$. In empirical applications with discrete estimators, such as the empirical distribution $\mathbb{P}_T$ that was used by [13] or the probability distributions implied by the generalized method of moments and generalized empirical likelihood that were used by [9], these methods and models are naturally redundant.

If the benchmark set K is continuous, then a problem reduction is achieved by substituting K for its extreme elements, $\mathrm{K}^{(0)}$. This substitution is harmless as a result of the convexity feature of the stochastic enhancement relation with respect to the host positions. It results in discretization if the number of extreme elements is finite, e.g., the vertices of a polyhedron.

Similarly, the utility class $\mathrm{cl}(\mathcal{U})$ may be substituted by its extreme elements, or $\mathcal{U}^{(0)}$ : $\mathrm{cl}(\mathcal{U}) = \mathrm{conv}\left(\mathcal{U}^{(0)}\right)$. This substitution is harmless because expected utility is a linear function of $u \in \mathrm{cl}(\mathcal{U})$, and $\mathrm{cl}(\mathcal{U})$ is a convex set. The substitution is useful because the extreme elements $u \in \mathcal{U}^{(0)}$ are low-dimensional functions for common specifications of $\mathcal{U}$, which simplifies the search over $\mathrm{cl}(\mathcal{U})$. Notably, for the general specification of $n$-th degree SD, $\mathcal{U}_n^{(0)}$ is comprised the one-parameter singularity functions $v_{n;\phi}(x) = -(\phi - x)_+^{n-1}$.

The total boundedness of the support, the Lipschitz continuity property of the positive part operator $(\cdot)_+$ and the compactness of $\Delta$ and $K$ imply that the one-dimensional parameter space

can be represented or approximated by a finite set with an arbitrary level of precision. For any $\epsilon > 0$, there exists a finite $\mathcal{U}_{n,\epsilon}^{(0)} \subseteq \mathcal{U}_n^{(0)}$ such that $\sup_{\kappa,\delta} \sup_{\phi \in \mathcal{U}^{(0)}} \inf_{\phi^\star \in \mathcal{U}_{n,\epsilon}^{(0)}} |D(u_{n,\phi}, \kappa, \delta, \mathbb{P}) - D(u_{n,\phi^\star}, \kappa, \delta, \mathbb{P})| < \epsilon$; $\mathcal{U}_{n,\epsilon}^{(0)}$ approximates $\mathcal{U}^{(0)}$ with precision $\epsilon$. Such a finite discretization is inconsequential for portfolio optimization for small enough values of the parameter.

Using the above replacements, system (3) can be reduced to

$$D(u, \kappa, \delta, \mathbb{P}) \geq 0 \; \forall (u, \kappa) \in \mathcal{U}^{(0)} \times \mathrm{K}^{(0)}; \tag{4}$$
$$\delta \in \Delta.$$

In what follows, it is assumed that $\mathrm{K}^{(0)}$ is finite; this assumption is justified by the fact that K is polyhedral in many applications. It is also assumed that, for any $\epsilon > 0$, there exists a finite $\mathcal{U}_\epsilon^{(0)}$ that approximates $\mathcal{U}^{(0)}$ with precision $\epsilon$. Since $\mathcal{U} \subseteq \mathcal{U}_2$, and $\mathcal{U}_2$ is the convex hull of $\mathcal{U}_2^{(0)}$, this assumption is harmless.

Under these assumptions, the semi-infinite system of inequalities (3) can be represented or approximated by the following finite system, given some choice of $\epsilon$:

$$D(u, \kappa, \delta, \mathbb{P}) \geq 0 \; \forall (u, \kappa) \in \mathcal{U}_\epsilon^{(0)} \times \mathrm{K}^{(0)}; \tag{5}$$
$$\delta \in \Delta.$$

In typical applications, a concave objective function $G(\delta, \mathbb{P})$ is maximized subject to system (5) for a given specification of $\mathcal{U}$ and K. As a result of the discretizations that are used, this optimization problem is finite and convex.

## 3. Empirical Counterparts

### 3.1. Consistency Features

We assume that the unobserved distribution function $\mathbb{P}$ is estimated by the empirical measure $\mathbb{P}_T$ from a sample of $T$ time-series observations $(x_t)_{t=1,\ldots,T}$. The payoffs generally are not stationary random variables because they include trend components. This statistical problem can be eliminated or mitigated by transforming the payoffs to gross investment returns $r := \mathrm{diag}^{-1}(p)x$ and transforming the asset holdings to portfolio weights $\lambda := c^{-1}(p \odot \kappa)$ (for hosts) and $\gamma := c^{-1}(p \odot \delta)$ (for overlays), which do not change the underlying investment decision-making problem. In order to isolate the effect of sampling variation on the stochastic enhancement restrictions, we also assume that the restrictions that define K and $\Delta$ are deterministic and thus are not affected by sampling error.

The following empirically weak SAO set is constructed:

$$\Delta_{(\mathcal{U},\mathrm{K},\mathbb{P}_T)}^{\mathrm{WAO}} = \{\delta \in \Delta : D(u, \kappa, \delta, \mathbb{P}_T) \geq 0 \; \forall (u, \kappa) \in \mathrm{cl}(\mathcal{U}) \times \mathrm{K}\} \tag{6}$$

This empirical counterpart of the latent set of population SAOs has several attractive statistical consistency features under three general assumptions that are stated below.

**Assumption 1.** *(Stationarity and Mixing). The process $(x_t)$ is stationary and absolutely regular. The asymptotic order of its mixing coefficients is $O(k^{-r})$ for some $r > 1$.*

These regularity conditions are compatible with stationary versions of ARMA-GARCH or stochastic volatility models, or more generally, processes that satisfy a general class of stochastic recurrence equations; see, for example, [23] (Section 6).

**Assumption 2.** *(Lipschitz Continuity). The population goal $G(\cdot, \mathbb{P})$ is Lipschitz continuous in $\delta$ with a Lipschitz coefficient that is continuous at $\mathbb{P}$ w.r.t. weak convergence.*

Given the boundedness of the support of $x_0$, Lipschitz continuity is easy to establish for standard goals such as expected return and expected utility. A uniform integrability

condition implied by the boundedness of the support ensures the required continuity of the coefficient in these cases. The objective function of [13] (the weighted average of individual appraisal ratios) also satisfies the assumption due to the CMT and the aforementioned uniform integrability; the Lipschitz coefficient is in this case a continuous function of moments, as long as the variance involved is strictly positive.

The final assumption generalizes the weak independence assumption of [9]. It is useful to avoid consistency problems that may arise for the empirical solutions due to binding inequalities (equivalences), or $D(u, \kappa, \delta, \mathcal{F}) = 0$, for some non-constant utility functions $u \in \mathrm{cl}(\mathcal{U})$.

**Assumption 3.** *(Joint Enhancement). For any non-constant $v \in cl(\mathcal{U})$, there exists some $\delta(v)$ that solves (3), such that $D(v, \kappa, \delta(v), \mathbb{P}) > 0$ for all $\kappa \in K$.*

The joint enhancement (henceforth, JE) assumption essentially ensures the existence of some neighborhood of every weak SAO $\delta_1 \in \Delta^{\mathrm{WAO}}_{(\mathcal{U},\mathrm{K},\mathbb{P})}$ that contains strict SAOs $\delta_2 \in \Delta^{\mathrm{SAO}}_{(\mathcal{U},\mathrm{K},\mathbb{P})}$, if strict SAOs exist. JE allows for different $\delta$ for different utility functions $v$ instead of a single $\delta$ for all such utilities. It mitigates the adverse effect of the binding inequalities on consistency. Given any weak SAO $\delta$ for which $D(v, \kappa, \delta, \mathbb{P}) = 0$ for some $(v, \kappa)$, consider the portfolio defined using the Lebesgue integral $\gamma := \int_{\mathcal{U}-\mathcal{U}^=} \delta(u) d((1-c)w_v + cw(u))$, $w \in \mathcal{W}$, where $\mathcal{W}$ is the set of strictly monotone Borel measures defined on $\mathcal{U}$, and $w_v$ is the degenerate measure at $v$, $c \in (0,1)$. JE implies that $D(u, \kappa, \gamma, \mathbb{P}) > 0$ for all $(u, \kappa) \in \mathcal{U} - \mathcal{U}^= \times \mathrm{K}$, and by choosing $c$ appropriately small, it can be chosen to lie as close to $\delta$ as desired.

Various consistency results are obtained based on the above assumptions. The limit theory evolves as $T \to \infty$. The squiggly arrow ($\rightsquigarrow$) represents convergence in distribution. The analysis also uses Painleve–Kuratowski (PK) convergence $\left(\overset{\mathrm{PK}}{\rightsquigarrow}\right)$ for sequences of compact subsets of $\Delta$. With high probability (w.h.p.) refers to probability converging to one. $\delta^*(\mathbb{P})$ represents the set of optimal weak SAOs, i.e., $\arg\max_{\Delta^{\mathrm{WAO}}_{(\mathcal{U},\mathrm{K},\mathbb{P})}} (G(\delta, \mathbb{P}))$. Given $\delta > 0$, $\Delta^{(\delta)}(\mathbb{P})$ represents the set of $\delta$-optimal weak SAOs, i.e., $\{\gamma \in \Delta^{\mathrm{WAO}}_{(\mathcal{U},\mathrm{K},\mathbb{P})} : \max_{\Delta^{\mathrm{WAO}}_{(\mathcal{U},\mathrm{K},\mathbb{P})}} G(\delta, \mathbb{P}) - G(\gamma, \mathbb{P}) \leq \delta\}$. $\Delta^=_{\mathbb{P}}$ denotes the set of trivial solutions $\left\{\delta \in \Delta : x^{\mathrm{T}}\delta = 0 \,\forall x : \mathbb{P}\left(\prod_{i=1}^N (-\infty, x_i]\right) > 0\right\}$; it obviously is a subset of the enlargement $\Delta^{\mathrm{WAO}}_{(\mathcal{U},\mathrm{K},\mathbb{P})} - \Delta^{\mathrm{SAO}}_{(\mathcal{U},\mathrm{K},\mathbb{P})}$.

**Theorem 1.** *Under Assumptions 1–3 the following results are obtained: (i) if $\Delta^{\mathrm{WAO}}_{(\mathcal{U},\mathrm{K},\mathbb{P})} - \Delta^=_{\mathbb{P}} \neq \varnothing$ and JE occurs, then for any $v \in \mathcal{U} - \mathcal{U}^=$ and for any $c_T \to 0$, such that $m_T c_T \to \infty$, we have that $\Delta^{\mathrm{WAO}}_{(\mathcal{U},\mathrm{K},\mathbb{P}_T)} \cap \Delta^{(c_T)}(\mathbb{P}) \overset{\mathrm{PK}}{\rightsquigarrow} \delta^*(\mathbb{P})$; (ii) if $\Delta^{\mathrm{WAO}}_{(\mathcal{U},\mathrm{K},\mathbb{P})} - \Delta^=_{\mathbb{P}} = \varnothing$, then $\Delta^{\mathrm{WAO}}_{(\mathcal{U},\mathrm{K},\mathbb{P}_T)} \overset{\mathrm{PK}}{\rightsquigarrow} \Delta^=_{\mathbb{P}}$.*

**Proof of Theorem 1.** Given that any element of $\Delta^=_{\mathbb{P}}$ belongs to $\Delta^{\mathrm{WAO}}_{(\mathcal{U},\mathrm{K},\mathbb{P})}$ and that when non-trivial weak SAOs do not exist, then $\Delta^{\mathrm{WAO}}_{(\mathcal{U},\mathrm{K},\mathbb{P})} = \Delta^=_{\mathbb{P}}$, if $\delta \notin \Delta^=_{\mathbb{P}}$, then $\exists u \in \mathcal{U}, \kappa \in K$ for which $D(u, \kappa, \delta, \mathbb{P}) < 0$. Assumption 1 and the compactness of $\mathcal{X}$, along with the FCLT of [24] (Corollary 4.1), imply that $\limsup_{T \to \infty} \mathbb{P}[D(u, \kappa, \delta, \mathbb{P}_T) \geq 0] = 0$, which establishes Theorem 1. (ii). Suppose next that non-trivial weak SAOs do not exist. Let $c_T = o(1)$ and $m_T c_T \to \infty$. Stationarity and mixing of $(x_t)$ and the compactness of $\mathcal{X}$, with corollary 2.E of [25], imply that we need to consider only fixed $u \in \mathcal{U}$ in our derivations. For any $\delta \in \delta^*(\mathbb{P})$ that does not have any trivial equivalences, we have that $\min_K D(u, \kappa, \delta, \mathbb{P}) > 0$ for all $u \in \mathcal{U} - \mathcal{U}^=$, thus $\liminf_{T \to \infty} \mathbb{P}[\min_K D(u, \kappa, \delta, \mathbb{P}_T) \geq 0] = 1$ for all $u \in \mathcal{U}$. Suppose then that $\delta$ has non-trivial equivalences. Then, for any $u \in \mathcal{U} - \mathcal{U}^=$ that does not correspond to some equivalence or to some $u \in \mathcal{U}^=$, the previous analysis holds. If $u \in \mathcal{U} - \mathcal{U}^=$ and $u$ corresponds to some non-trivial equivalence, then, for a large enough $T$, consider the strong SAO $\gamma_T := \int_{\mathcal{U}-\mathcal{U}^=} \delta(u) d((1-c_T^\star)w_v + c_T^\star w(u))$ for $L$ the Lipschitz coefficient of $u$. Here, $c_T^\star = \frac{c_T}{L \mathrm{diam}(\Delta)} = O(c_T)$. Notice that since (i) $\min_K D(u, \kappa, \cdot, \mathbb{P})$ is concave, (ii) by

construction $\min_K D(u, \kappa, \delta, \mathbb{P}) = 0$ and (iii) $D\left(u, \kappa, \int_{\mathcal{U}-\mathcal{U}^=} \delta(u) dw(u), \mathbb{P}\right) > 0$ due to joint enhancement, we have that

$$-m_T \inf_K D(u, \kappa, \gamma_T, \mathbb{P}) \leq -m_T c_T^\star \inf_K D\left(u, \kappa, \int_{\mathcal{U}-\mathcal{U}^=} \delta(u) dw(u), \mathbb{P}\right) < 0,$$

and thereby

$$\mathbb{P}[m_T \inf_K D(u, \kappa, \gamma_T, \mathbb{P}_T - \mathbb{P}) \geq -m_T \inf_K D(u, \kappa, \gamma_T, \mathbb{P})]$$
$$\geq \mathbb{P}\left[m_T \inf_K D(u, \kappa, \gamma_T, \mathbb{P}_T - \mathbb{P}) \geq -m_T c_T^\star \inf_K D\left(u, \kappa, \int_{\mathcal{U}-\mathcal{U}^=} \delta(u) dw(u), \mathbb{P}\right)\right],$$

and, due to Assumption 1, the CMT and the Portmanteau Theorem, the lim inf of the latter probability is greater than or equal to $\mathbb{P}[\min_K \mathcal{G}(u, \kappa, \delta) \geq -\infty] = 1$. Hence, we have that $\liminf_{T \to \infty} \mathbb{P}\left[\gamma_T \in \Delta^{\mathrm{WAO}}_{(\mathcal{U},\mathrm{K},\mathbb{P}_T)}\right] = 1$ and $\gamma_T \to \delta$. The previous then imply that, in all cases, $\delta$ lies in the lim inf of $\Delta^{\mathrm{WAO}}_{(\mathcal{U},\mathrm{K},\mathbb{P}_T)}$ w.h.p. Now, due to the definitions of $\delta$, $\gamma_T$ and the Lipschitz continuity of $G(\delta, \mathbb{P})$, we obtain

$$0 \leq G(\delta, \mathbb{P}) - G(\gamma_T, \mathbb{P})$$
$$\leq \max_K Lc_T^\star \left\| \delta - \int_{\mathcal{U}-\mathcal{U}^=} \delta(u) dw(u) \right\| \leq c_T^\star L \mathrm{diam}(\Delta) = c_T, \tag{7}$$

where the first inequality follows from (7). This implies that $\gamma_T \in \Delta^{(c_T)}_{(\mathcal{U},\mathrm{K})}(v, \mathbb{P})$. Obviously $\delta^*(\mathbb{P}) \subseteq \Delta^{(c_T)}(\mathbb{P})$. Any other element of $\Delta^{(c_T)}(\mathbb{P})$ that lies in the empirical weak SAO set with asymptotically positive probability will necessarily converge to some element of $\delta^*(\mathbb{P})$. The previous establish Theorem 1. (i), since there cannot exist accumulation points of sequences of elements of $\Delta^{(c_T)}(\mathbb{P})$ that lie outside $\delta^*(\mathbb{P})$. $\quad\square$

The following result complements Theorem 1 by showing that the empirical optimal solutions also approximate the original as $T \to \infty$.

**Theorem 2.** *(Empirical Solution Properties). Under the premises of Theorem 1, the following results are obtained: (i)*

$$\sup_{\delta \in \Delta^{\mathrm{WAO}}_{(\mathcal{U},\mathrm{K},\mathbb{P}_T)}} G(\delta, \mathbb{P}_T) \rightsquigarrow \sup_{\delta \in \Delta^0_{(\mathcal{U},\mathrm{K},\mathbb{P})}} G(\delta, \mathbb{P}), \tag{8}$$

*where $\Delta^0_{(\mathcal{U},\mathrm{K},\mathbb{P})} := \delta^*(\mathbb{P})$ if $\Delta^{\mathrm{WAO}}_{(\mathcal{U},\mathrm{K},\mathbb{P})} - \Delta^=_{\mathbb{P}} \neq \varnothing$ and joint enhancement (7) occurs, and $\Delta^0_{(\mathcal{U},\mathrm{K},\mathbb{P})} := \Delta^=_{\mathbb{P}}$ if $\Delta^{\mathrm{WAO}}_{(\mathcal{U},\mathrm{K},\mathbb{P})} - \Delta^=_{\mathbb{P}} = \varnothing$; (ii) if $\Delta^{\mathrm{WAO}}_{(\mathcal{U},\mathrm{K},\mathbb{P})} - \Delta^=_{\mathbb{P}} \neq \varnothing$ and JE occurs, then every limit of any subsequence of elements of $\delta^*(\mathbb{P}_T)$ lies in $\delta^*(\mathbb{P})$; (iii) if $\Delta^{\mathrm{WAO}}_{(\mathcal{U},\mathrm{K},\mathbb{P})} - \Delta^=_{\mathbb{P}} = \varnothing$, then $\delta^*(\mathbb{P}_T) \overset{PK}{\rightsquigarrow} \Delta^=_{\mathbb{P}}$.*

**Proof of Theorem 2.** Assumption 1, the compactness of $\mathcal{X}$, along with the FCLT of [24] (Corollary 4.1), the concavity of $G(\delta, \mathcal{F}_T)$ and Skorokhod representations applicable due to Theorem 3.7.25 of [26] imply the w.h.p. epi-convergence of the latter to $G(\delta, \mathcal{F})$ due to corollary 2.E of [25]. Then, the results in (i), (ii) and (iii) follow from Theorem 1, employing Skorokhod representations applicable due to Theorem 3.7.25 of [26], using Proposition 3.2, in Chapter 5 of [27] and then reverting to the original probability space. Specifically for (iii), and since, $\mathbb{E}_{\mathbb{P}_T}\left[u(x^{\mathrm{T}}(\kappa + \delta))\right] - \mathbb{E}_{\mathbb{P}_T}\left[u(x^{\mathrm{T}}\kappa)\right] \leq 0$ w.h.p., for any $\delta \in \Delta$, $u \in \mathcal{U}, \kappa \in K$, that due to Assumption 2, $G(\delta, \mathbb{P}_T)$ is constant on $\Delta^=_{\mathbb{P}}$ w.h.p., it follows that any $\delta \notin \Delta^=_{\mathbb{P}}$ will not lie in $\Delta^{\mathrm{WAO}}_{(\mathcal{U},\mathrm{K},\mathbb{P}_T)}$ and thereby in $\delta^*(\mathbb{P}_T)$ w.h.p., any $\delta \in \Delta^=_{\mathbb{P}}$ lies in $\Delta^{\mathrm{WAO}}_{(\mathcal{U},\mathrm{K},\mathbb{P}_T)}$ a.s. for all $T$, and due to the constancy of $G$, it also lies in $\delta^*(\mathbb{P}_T)$ w.h.p. $\quad\square$

The above consistency properties imply that the probability of each of two possible types of decision errors tends to zero: (I) the selected portfolio is an empirical SAO but not a population SAO; (II) the selected portfolio is optimal under the empirical measure but

suboptimal under the population distribution because the population optimum is not an empirical SAO.

For a type I error, the following reasoning applies: if $\delta \notin \Delta^{\text{WAO}}_{(\mathcal{U},\text{K},\mathbb{P})}$, then there exists at least one strict inequality $D(u, \kappa, \delta, \mathbb{P}) < 0$, for some $(u, \kappa) \in \mathcal{U} \times \text{K}$. Since strict inequalities are asymptotically unaffected by sampling error in our stationary and ergodic framework, the probability that such a $\delta$ is falsely included in the empirical SAO set is asymptotically negligible.

As far as a type II error is concerned, there is a non-vanishing probability of false exclusion of non-strict SAOs $\delta \in \left( \Delta^{\text{WAO}}_{(\mathcal{U},\text{K},\mathbb{P})} - \Delta^{\text{SAO}}_{(\mathcal{U},\text{K},\mathbb{P})} \right)$, which feature contacts $D(u, \kappa, \delta, \mathbb{P}) = 0$ for some $(u, \kappa) \in \mathcal{U} \times \text{K}$. However, the neighborhood of these non-robust solutions includes strict SAOs $\delta_T \in \Delta^{\text{SAO}}_{(\mathcal{U},\text{K},\mathbb{P})}$ that feature asymptotically strict (scaled) empirical inequalities $\sqrt{T}D(u, \kappa, \delta_T, \mathbb{P}_T) > 0$ for all $(u, \kappa) \in \mathcal{U} \times \text{K}$, and that are therefore included in $\Delta^{\text{WAO}}_{(\mathcal{U},\text{K},\mathbb{P}_T)}$ with high probability.

When $\mathcal{U}^{(0)}_\epsilon$ and $\text{K}^{(0)}$ are used in place of $\mathcal{U}$ and K, the results above still hold as long as $\epsilon \to 0$ and $T \to \infty$, and therefore $\mathcal{U}^{(0)}_\epsilon$ converges to $\mathcal{U}^{(0)}$ in the Painleve–Kuratowski topology. This is true even in the case where the latter convergence is in probability, e.g., whenever consistent estimators for a potentially latent upper bound of the support are used in the discretization.

### 3.2. Empirical Likelihood Ratio Test for Being an SAO

In their empirical analysis, ref. [13] construct arbitrage portfolios that are SAOs in a given sample and test whether these are population SAOs out of the sample. They thus evaluate a single, optimized overlay with portfolio weights that are fixed and known out of sample, thereby avoiding distortions of the test size stemming from overfitting the data.

In this context, the obvious choice of the null hypothesis seems to be $\mathbf{H}_0 : \delta \in \Delta^{\text{WAO}}_{(\mathcal{U},\text{K},\mathbb{P})}$. This specification is further supported by the theoretical prediction that a portfolio that is optimized subject to empirical SAO restrictions converges to a true population SAO, if SAOs exist. Therefore, we discuss the testing of the null $\mathbf{H}_0$ before discussing the testing of the alternative $\mathbf{H}_1 : \delta \notin \Delta^{\text{SAO}}_{(\mathcal{U},\text{K},\mathbb{P})}$ in the second part of this section.

The use of $\mathbf{H}_0$ is also consistent with the usual practice for statistical tests of pairwise dominance. Standard tests such as those of [28] focus on null dominance, due to limiting degeneracies under the alternative of non-dominance. The current null provides a generalization of the standard specification to multiple pairwise dominance relations between combined portfolios and the underlying hosts.

Standard tests for pairwise dominance are usually based on Kolmogorov–Smirnov or Cramer–von Mises statistics for measuring violations of the empirical moment inequalities. These statistics are analytically convenient due to their computational tractability for lattice distributions and their straightforward compatibility with tools of asymptotic analysis such as Glivenko–Cantelli and Donsker theorems for uniform convergence, as well as generalized delta methods. Rejection regions are usually approximated by re-sampling methods that allow for asymptotically exact and consistent inference under stationarity and mixing.

Unfortunately, the Kolmogorov–Smirnov and Cramer–von Mises statistics are relatively inefficient for financial data sets with short time series of non-overlapping, low-frequency returns and broad cross-sections. Re-sampling can also be computationally costly if large optimization problems need to be solved for thousands of pseudo-samples or sub-samples.

To enhance statistical efficiency in finite samples and reduce computer burden, an alternative approach is adopted here based on BEL. A similar approach is used in the tests for non-dominance, efficiency and optimality by [18–20]. The proposed approach generalizes these earlier studies by allowing for multiple host portfolios, testing both the null and the alternative, and generalizing the data dependence structure.

A prominent role in our analysis is played by the 'contact set', or set of the binding inequalities, $\mathrm{CS}_0 := \left\{ (u, \kappa) \in \mathcal{U}^{(0)} \times \mathrm{K}^{(0)} : D(u, \kappa, \delta, \mathbb{P}) = 0 \right\}$. Its cardinality is represented by $N_0(\delta, \mathbb{P})$; this number is crucial for the approximation of the rejection region of the testing procedure. The analysis below describes sufficient conditions for finiteness of $\mathrm{CS}_0$ for every member of $\Delta^{\mathrm{WAO}}_{(\mathcal{U}, \mathrm{K}, \mathbb{P})}$ as well as a modification of the procedure that accounts for the possibility of an infinite contact set.

To account for temporal dependence, the sample is subdivided into $T^* := \lfloor T - B / L \rfloor + 1$ potentially overlapping blocks of $B$ consecutive observations, $\mathcal{B}_s := \{ \boldsymbol{x}_{(s-1)L+1}, \cdots, \boldsymbol{x}_{(s-1)L+B} \}$, $s = 1, \cdots, T^*$, with $L \leq B$. $L$ is considered to be independent of $T$, and $B$ is assumed to diverge with a rate slower than $\sqrt{T}$. The optimal choice of the block size $B$ is case-dependent and usually involves of a trade-off between the data dynamics and the number of asymptotically independent blocks.

In the following, $\mathcal{G}_T$ denotes the set of probability distributions on the set of blocks, and $\mathbb{G}_T \in \mathcal{G}_T$ denotes the empirical measure, i.e., $\mathbb{G}_T(\mathcal{B}) := (T^*)^{-1} \sum_{s=1}^{T^*} \mathbb{I}[\mathcal{B}_s = \mathcal{B}]$ for any block $\mathcal{B}$. The test statistic measures the smallest possible adjustments to the probability mass function of $\mathbb{G}_T$ that ensure that the evaluated overlay is a weak SAO; the optimality property is w.r.t. the divergence from the adjusted distribution to the empirical measure of the blocks. The BELR test statistic can be computed based on a solution to the following minimum relative entropy (MRE) problem:

$$\min_{\mathbb{G} \in \mathcal{G}_T} \mathrm{KL}(\mathbb{G}_T \| \mathbb{G}) \tag{9}$$

$$\text{s.t. } \delta \in \Delta^{\mathrm{WAO}}_{(\mathcal{U}, \mathrm{K}, \mathbb{G})}.$$

In this expression, KL denotes the Kullback–Leibler divergence: $\sum_{s=1}^{T^*} \mathbb{G}_T(\mathcal{B}_s) \ln(\mathbb{G}_T(\mathcal{B}_s) / \mathbb{G}(\mathcal{B}_s))$. This measure gives a well-known information-theoretic representation of the dissimilarity between two measures sharing a common support (see [29]). For $\epsilon > 0$, the finite system (5) is used so that the MRE problem (9) is approximated by an optimization problem that involves a finite system of moment inequalities:

Since any $\mathbb{G} \in \mathcal{G}_T$ is discrete, only the probability mass levels at the blocks $\mathbb{G}(\mathcal{B}_s)$, $s = 1, \cdots, T^*$, need to be determined, and the minimum relative entropy problem reduces to a convex optimization problem, given the formulation in system (4):

$$\min_{\mathbb{G} \in \mathcal{G}_T} \mathrm{KL}(\mathbb{G}_T \| \mathbb{G}) \tag{10}$$

$$\text{s.t. } D(u, \kappa, \delta, \mathbb{G}) \geq 0 \ \forall (u, \kappa) \in \mathcal{U}^{(0)}_\epsilon \times \mathrm{K}^{(0)},$$

where $D(u, \kappa, \delta, g) := \sum_{s=1}^{T^*} \mathbb{G}(s) \left( \frac{1}{B} \sum_{j=1}^{B} u(\boldsymbol{x}^{\mathrm{T}}_{(s-1)L+j}(\kappa + \delta)) - u(\boldsymbol{x}^{\mathrm{T}}_{(s-1)L+j} \kappa) \right)$.

Inference on $\mathbf{H}_0$ can be based on the BELR test statistic $\mathrm{ELR}_T = 2 \frac{T}{T^* B} \cdot \mathrm{KL}(\mathbb{G}_T \| \mathbb{G}(\delta))$, where $\mathbb{G}(\delta)$ is a solution of the variational problem (9). The exact limit null distribution of ELR is a chi-bar-squared distribution under the aforementioned assumptions about the data and the blocks. This null distribution is not directly implementable in rejection region analysis because its mixing weights depend on the latent $\mathbb{P}$. An asymptotically conservative test can, however, be obtained via majorizing chi-squared distributions and methods of moment selection.

Specifically, the limiting null distribution is dominated by $\chi^2_{N_0(\delta, \mathbb{P})}$, the degrees of freedom of which equal the number of binding inequalities (the cardinality of the contact set) whenever finite. The number of contacts can be consistently estimated by the number of empirical moment conditions that are approximately binding, or $N_0(\delta, \mathbb{G}_T, c_T) := \mathrm{card} \left\{ (u, \kappa) \in \mathcal{U}^{(0)}_\epsilon \times \mathrm{K}^{(0)} : |D(u, \kappa, \delta, \mathbb{G}_T)| \leq c_T \right\}$, where the slack $c_T > 0$ is a potentially degenerate random variable that weakly converges to zero at an appropriate rate.

Consequently, an asymptotically conservative rejection region can be formed using the stochastic distribution $\chi^2_{N_0(\delta,\mathbb{G}_T,c_T)}$, so that the test size, or the probability of false rejection of an SAO, is asymptotically less than or equal to the nominal significance level.

The following result derives the limit theory of the testing procedure defined by (10) and the rejection of $\mathbf{H}_0$ if $\mathrm{ELR}_T \geq q\left(1 - \alpha, \chi^2_{N_0(\delta,\mathbb{P}_T,c_T)}\right)$, where $q\left(1 - \alpha, \chi^2_{N_0(\delta,\mathbb{P}_T,c_T)}\right)$ represents the $1 - \alpha$ quantile of the stochastic distribution $\chi^2_{N_0(\delta,\mathbb{P}_T,c_T)}$ for $\alpha \in (0,1)$. The result shows asymptotic conservatism and consistency.

**Theorem 3.** *Suppose that Assumption 1 holds, and (a) CS is finite, (b) there exists some $\varepsilon > 0$ such that,*

$$\lambda_{\min}(\mathcal{V}) > \varepsilon,$$

*where $\lambda_{\min}(\mathcal{V})$ represents the minimum eigenvalue of*

$$\mathcal{V} := \mathbb{E}\left[\left(u\left(x_0^{\mathrm{T}}(\kappa + \delta)\right) - u\left(x_0^{\mathrm{T}}\kappa\right)\right)\left(u^{\star}\left(x_0^{\mathrm{T}}(\kappa^{\star} + \delta)\right) - u^{\star}\left(x_0^{\mathrm{T}}\kappa^{\star}\right)\right)^{\mathrm{T}}\right]_{(u,\kappa),(u^{\star},\kappa^{\star})\in\mathrm{CS}_0},$$

*(c) the tolerance parameter satisfies $c_T \to 0$, while $\sqrt{T}c_T \to +\infty$ almost surely, (d) the block size satisfies $B \to +\infty$ and $B = O(T^{\rho})$ for $0 < \rho < \frac{1}{2}$ and $L$ is independent of $T$, and (e) $\epsilon \to 0$ with $T$, so that $\mathcal{U}_{\epsilon}^{(0)}$ converges to to a non-stochastic dense subset of $\mathcal{U}^{(0)}$ in probability, and $\mathrm{card}\left(\mathcal{U}_{\epsilon}^{(0)}\right) = o_p(\frac{T}{B})$.*

*Then, the following results are obtained: (i)*

$$\mathrm{ELR}_T \rightsquigarrow \begin{cases} \inf_{v\in\mathbb{R}_{+}^{N_0(\delta,\mathbb{P})}} (\mathbb{C} - v)^{\mathrm{T}}\mathcal{V}_{\mathbb{C}}^{-1}(\mathbb{C} - v), & \mathbf{H}_0 \wedge N_0(\delta,\mathbb{P}) \neq 0 \\ 0, & \mathbf{H}_0 \wedge N_0(\delta,\mathbb{P}) = 0, \\ +\infty, & \mathbf{H}_1 \end{cases} \tag{11}$$

*where $\mathbb{C}$ is a zero-mean Gaussian vector with covariance matrix*

$$\mathcal{V}_{\mathbb{C}} := \mathcal{V} + 2\sum_{t=1}^{\infty} \mathbb{E}\left[\left(u\left(x_0^{\mathrm{T}}(\kappa + \delta)\right) - u\left(x_0^{\mathrm{T}}\kappa\right)\right)\left(u^{\star}\left(x_t^{\mathrm{T}}(\kappa^{\star} + \delta)\right) - u^{\star}\left(x_t^{\mathrm{T}}\kappa^{\star}\right)\right)^{\mathrm{T}}\right]_{(u,\kappa),(u^{\star},\kappa^{\star})\in\mathrm{CS}};$$

*(ii) under $\mathbf{H}_0 \wedge N_0(\delta,\mathbb{P}) \neq 0$,*

$$\limsup_{T\to\infty} \mathbb{P}\left(\mathrm{ELR}_T \geq q\left(1 - \alpha, \chi^2_{N_0(\delta,\mathbb{P}_T,c_T)}\right)\right) \leq \alpha, \tag{12}$$

*while under $\mathbf{H}_0 \wedge N_0(\delta,\mathbb{P}) = 0$,*

$$\lim_{T\to\infty} \mathbb{P}\left(\mathrm{ELR}_T \geq q\left(1 - \alpha, \chi^2_{N_0(\delta,\mathbb{P}_T,c_T)}\right)\right) = 0; \tag{13}$$

*(iii) under $\mathbf{H}_1$,*

$$\lim_{T\to\infty} \mathbb{P}\left(\mathrm{ELR}_T \geq q\left(1 - \alpha, \chi^2_{N_0(\delta,\mathbb{P}_T,c_T)}\right)\right) = 1. \tag{14}$$

**Proof of Theorem 3.** Equation (11) follows as in the proof of Theorem 4.2.1 of [30]. Consider the case $\mathrm{H}_0 \wedge N_0(\delta,\mathbb{P}) \neq 0$. Uniformly w.r.t. the $(i,j) \notin \mathrm{CS}_0$, it is found that, due to the definition of the tolerance parameter and the Birkhoff's ULLN, $\mathbb{E}_{\mathbb{G}_T}\left[u\left(x_0^{\mathrm{T}}(\kappa + \delta)\right) - u\left(x_0^{\mathrm{T}}\kappa\right)\right] > c_T$, eventually, almost surely. Using Skorokhod representations, since $\sqrt{T}c_T$ diverges to infinity almost surely, uniformly over the set $\mathrm{CS}_0$, then $\left|\sqrt{T}\mathbb{E}_{\mathbb{G}_T}\left[u\left(x_0^{\mathrm{T}}(\kappa + \delta)\right) - u\left(x_0^{\mathrm{T}}\kappa\right)\right]\right| \leq \sqrt{T}c_T$, eventually, almost surely. The previous along with (e) imply that $N_0(\delta,\mathbb{P}_T,c_T) \rightsquigarrow N_0(\delta,\mathbb{P})$, jointly with $\mathrm{ELR}_T$, and thereby we obtain that

$$\mathrm{ELR}_T \rightsquigarrow \inf_{v \in \mathbb{R}_+^{N(\delta,\mathbb{P})}} (\mathbb{C} - v)^{\mathrm{T}} \mathcal{V}_{\mathbb{C}}^{-1} (\mathbb{C} - v)$$

$$= \inf_{v \in \mathbb{R}_+^{N(\delta,\mathcal{F})}} \left[ \begin{array}{c} \mathbb{C}^{\mathrm{T}} \mathcal{V}_{\mathbb{C}}^{-1} \mathbb{C} \\ - \inf_{v \in C^o} (\mathbb{C} - v)^{\mathrm{T}} \mathcal{V}_{\mathbb{C}}^{-1} (\mathbb{C} - v) \end{array} \right] \le \mathbb{C}^{\mathrm{T}} \mathcal{V}_{\mathbb{C}}^{-1} \mathbb{C},$$

due to Proposition 3.4.1 of [31], where $C^o$ represents the polar cone of $\mathbb{R}_+^{N_0(\delta,\mathbb{P})}$. Then, the Portmanteau Theorem establishes (12). For the case $\mathbf{H}_0 \wedge N_0(\delta,\mathbb{P}) = 0$, (11) implies that $\mathrm{ELR}_T$ is eventually zero w.h.p., hence (13) follows. Furthermore, under the alternative, the proof of Theorem 4.2.1 of [30] implies that $\mathrm{ELR}_T \ge O_p(\frac{T}{B})$. Thereby, the growth condition on the approximation of $\mathcal{U}$ along with (c), and via the use of Skorokhod representations, imply that the modified statistic diverges to infinity, while the quantile $q\left(1 - \alpha, \chi^2_{N_0(\delta,\mathbb{P}_T,c_T)}\right)$ is almost surely bounded, hence (14) follows. $\square$

The test is consistent under the alternative hypothesis due to the divergence to infinity of the test statistic and the boundedness from above of the quantiles used as critical values. However, the test is asymptotically conservative, due to the majorizing properties of the critical values and the restricted limiting behavior of the slacks.

For any $\delta \in \Delta_{(\mathcal{U},\mathrm{K},\mathbb{P})}^{\mathrm{WAO}}$ that does not belong to the generic set of SAOs with finite contacts discussed above, the contact set finiteness condition (a) holds whenever $D(\cdot, \kappa_j, \delta, \mathbb{P})$ is analytic in the Russell–Seo threshold parameter for every extreme point $\kappa_j$. Analyticity would be in turn obtained if $\mathbb{P}$ has an analytic density, due to our bounded support framework.

Under the weaker assumption of a continuously differentiable density, another path for obtaining (a) is to ensure that the Hessian of $D(u_{n,\phi}, \kappa_i, \delta, \mathbb{P})$ w.r.t. $\phi$ has a finite number of zeros. For example, when $\mathcal{U} = \mathcal{U}_2$ and $\mathcal{U}^{(0)}$ is the set of Russell–Seo elementary utilities, using [32], we obtain that the Hessian equals $\mathbb{P}(\{\phi = (\kappa_i + \delta)^{\mathrm{T}} x\}) - \mathbb{P}(\{\phi = \kappa_i^{\mathrm{T}} x\})$. Using then the diffeo-geometric analysis in Paragraph 5 of [33], we obtain that the Hessian is not zero whenever $\delta^{\mathrm{T}} x$ is of the same sign for almost every value of $x$ on the boundary of $\{\phi = \kappa_i^{\mathrm{T}} x\}$ locally uniformly in the null hypothesis.

The strictly positive minimum eigenvalue condition (b) is necessary for sequential convergence of the test statistic under the null hypothesis. Given (a) and if trivial contacts with zero empirical variance are excluded from the analysis, this condition holds whenever the random vector $\left(u(x_0^{\mathrm{T}}(\kappa, \delta)) - u(x_0^{\mathrm{T}} \kappa)\right)_{\mathrm{CS}_0}$ has a full rank covariance matrix; this can, for example, be tested via the characteristic roots-based tests of [34] using the empirical covariance matrix of the non-trivial empirical contacts.

The restriction of the asymptotic behavior of the tolerance parameter (c) is usual in the econometric literature; see [35] and references therein. The restriction on the block size divergence rate (d) is also standard (see, for example, Theorem 3 of [36]).

The current first-order limit theory is silent about the optimal choice of the block size $B$ beyond specifying general rates of convergence. Finer details could be obtained by higher-order asymptotics, and it is expected that optimality depends crucially on the temporal-dependence structure of the underlying returns' process (see, for example, Section 2 of [37]). The optimal $B$ can be approximated by some empirical variance minimization method (see, for example, [38]).

For the important case of $n$-th degree SD, or $\mathcal{U}^0 = \mathcal{U}_n^0$, condition (e) is satisfied when, e.g., the empirical support is partitioned into $\lfloor (T/B)^\alpha \rfloor$ sub-intervals, for some $\alpha < 1$, and the Russell–Seo thresholds are placed at the boundaries of those sub-intervals.

Whenever the CS is infinite and (a) fails, a simple modification of the test statistic via the estimated number of contacts $N_0(\delta, \mathbb{P}_T, c_T)$ would lead to a standard normal limiting null bound distribution. Specifically, performing the test based on the rejection rule

$$\left(\mathrm{ELR}_T - N_0(\delta, \mathbb{P}_T, c_T)\right) / \sqrt{2 N_0(\delta, \mathbb{P}_T, c_T)} > \left(q(1 - \alpha, \chi^2_{N_0(\delta, \mathbb{P}_T, c_T)}) - N_0(\delta, \mathbb{P}_T, c_T)\right) / \sqrt{2 N_0(\delta, \mathbb{P}_T, c_T)}$$

can be proven to satisfy (ii) and (iii) of the previous result, if $V$ has a spectrum bounded away from zero, (c), (d) and (e) hold. This is obvious whenever $\mathrm{CS}_0$ is finite from Theorem 3.

Whenever $CS_0$ is infinite, then the spectrum condition and the fact that $\mathcal{U}_\epsilon^{(0)}$ is finite for any $T$, imply that, under the null hypothesis, the modified statistic will be bounded above by a standard normal distribution. The lhs of the rejection rule can be shown to converge in distribution to the $1 - \alpha$ quantiles of the standard normal. Furthermore, under the alternative, the proof of Theorem 4.2.1 of [30] implies that $ELR_T \geq O_p(\frac{T}{B})$. Thereby, the growth condition on the approximation of $\mathcal{U}$ along with (c), and via the use of Skorokhod representations, imply that the modified statistic diverges to infinity. Hence, under the particular restriction on the growth rate of $\mathcal{U}_\epsilon^{(0)}$, we obtain a robust modification of the original test that avoids (a).

In cases where $cl(\mathcal{U})$ contains utilities that correspond to contacts with zero empirical variance, the spectrum boundedness away from zero condition can have a restricted stochastic dominance interpretation. For example, when $cl(\mathcal{U})$ is the set of Russell-Seo utilities-see [21], then the fact that the population support infimum $a$ is finite implies that, for the condition to hold, a set of thresholds to the right of $a$ should be excluded from the analysis. The analyst can exclude the thresholds in the interval $[a_T, a_T + \delta]$, where $a_T$ is the empirical support infimum and $\delta$ is sufficiently small to ensure that the excluded Russell–Seo utilities are inconsequential, and then test whether the condition holds for the remaining utilities using the [39] rank tests on the empirical covariance of the empirical contacts.

### 3.3. Testing the Alternative of Not Being an SAO

The hypothesis $\mathbf{H}_0$ generally allows for the construction of tests that are locally more powerful in fixed samples than the alternative $\mathbf{H}_1 : \delta \notin \Delta_{(\mathcal{U},K,\mathbb{P})}^{WAO}$ or $\mathbf{H}_1 : \delta \notin \Delta_{(\mathcal{U},K,\mathbb{P})}^{SAO}$ because data variation is generally more likely to yield false non-dominance classifications than false dominance classifications; the existence of a single inequality with the 'wrong' sign suffices for the system of dominance inequalities to be violated. Nevertheless, the conservative nature of the proposed test for $\mathbf{H}_0$ could compromise power at the boundary between the two hypotheses; a possible local lack of power of the BELR test for the null could be mitigated using a second test that evaluates the alternative.

For this purpose, the analysis is completed here with the design of an ELR test for the alternative hypothesis $\mathbf{H}_1 : \delta \notin \Delta_{(\mathcal{U},K,\mathbb{P})}^{SAO}$. For finite $\mathcal{U}_\epsilon^{(0)}$ and $K^{(0)}$, the condition $\delta \in \Delta_{(\mathcal{U},K,\mathbb{P})}^{SAO}$ amounts to a finite system: $D(u,\kappa,\delta,\mathbb{P}) > 0, \forall (u,\kappa) \in \mathcal{U}_\epsilon^{(0)} \times K^{(0)}$. The condition $\delta \notin \Delta_{(\mathcal{U},K,\mathbb{P})}^{SAO}$ requires violations of this system. Violations occur whenever $\delta \in \Delta_{(\mathcal{U},K,\mathbb{P})}^{WAO} - \Delta_{(\mathcal{U},K,\mathbb{P})}^{SAO}$ or when $\delta \notin \Delta_{(\mathcal{U},K,\mathbb{P})}^{WAO}$. In the latter case, violations would imply asymptotic non-tightness for Kolmogorov–Smirnov or Cramer–von Mises type statistics under the alternative. Violations can, however, be additionally characterized as solutions to the alternative system $\sum_{u \in \mathcal{U}_\epsilon^{(0)}} \sum_{\kappa \in K^{(0)}} \beta_{u,\kappa} D(u,\kappa,\delta,\mathbb{P}) \leq 0$; $\sum_{u \in \mathcal{U}_\epsilon^{(0)}} \sum_{\kappa \in K^{(0)}} \beta_{u,\kappa} = 1$; $\beta_{u,\kappa} \geq 0 \, \forall (u,\kappa) \in \mathcal{U}_\epsilon^{(0)} \times K^{(0)}$. Non-trivial binding inequalities for the alternative system always exist, and this implies tightness for the testing procedure that is described below. Using the block structure above, the relevant relative entropy problem follows:

$$\min_{\mathbb{G} \in \mathcal{G}_T} KL(\mathbb{G}_T \| \mathbb{G}) \tag{15}$$

$$\text{s.t.} \sum_{u \in \mathcal{U}_\epsilon^{(0)}} \sum_{\kappa \in K^{(0)}} \beta_{u,\kappa} D(u,\kappa,\delta,\mathbb{G}) \leq 0;$$

$$\sum_{u \in \mathcal{U}_\epsilon^{(0)}} \sum_{\kappa \in K^{(0)}} \beta_{u,\kappa} = 1; \beta_{u,\kappa} \geq 0; \forall (u,\kappa) \in \mathcal{U}_\epsilon^{(0)} \times K^{(0)}.$$

This problem is bi-linear in $(\beta, \mathbb{G})$ and hence bi-convex. It can be solved using an alternating direction method of moments (ADMM) algorithm that alternates between optimizing w.r.t. $\beta$ and optimizing w.r.t. $\mathbb{G}$.

The contact set associated with testing the alternative hypothesis is defined as $CS_1 := \left\{ \beta : \sum_{u \in \mathcal{U}_\epsilon^{(0)}} \sum_{\kappa \in K^{(0)}} \beta_{u,\kappa} D(u,\kappa,\delta,\mathbb{P}) = 0 \right\}$. As noted above, this is non-empty

even in cases where $\mathcal{U} = \mathrm{cl}(\mathcal{U})$ and no utilities associated with trivial contacts of zero empirical variance appear in the analysis. For testing the alternative, non-finiteness of the contact set is more common than for testing the null; the alternative hypothesis imposes moment inequalities of differing signs and may also contain binding moment inequalities. In this case, there exist infinite convex combinations of the inequalities involved that produce zeros. Thus, inference is performed by the modification of the BELR statistic via translation and scaling by the number of empirical contacts and the finite approximation of the simplex in which parameter $\boldsymbol{\beta}$ attains its values. The following rejection rule used is

$$(\mathrm{ELR}_T^\star - N_1(\delta, \mathbb{P}_T, c_T)) / \sqrt{2N_1(\delta, \mathbb{P}_T, c_T)} > \left( q(1-\alpha, \chi^2_{N_1(\delta, \mathbb{P}_T, c_T)}) - N_1(\delta, \mathbb{P}_T, c_T) \right) / \sqrt{2N_1(\delta, \mathbb{P}_T, c_T)},$$

where $\mathrm{ELR}_T^\star = 2\frac{T}{T^*B} \cdot \mathrm{KL}(\mathbb{G}_T \| \mathbb{G}^\star(\delta))$, with $\mathbb{G}^\star(\delta)$ the solution of (15), and, similarly to the previous, $N_1(\delta, \mathbb{P}_T, c_T) := \mathrm{card}\left\{ \boldsymbol{\beta} \in S_T : \left| \sum_{u \in \mathcal{U}_\epsilon^{(0)}} \sum_{\kappa \in \mathrm{K}^{(0)}} \beta_{u,\kappa} D(u, \kappa, \delta, \mathbb{G}_T) \right| \le c_T \right\}$ is the number of empirical contacts, and $S_T$ a (potentially stochastic) finite discretization of the $(\#(\mathcal{U}_\epsilon^{(0)} \times K^0) - 1)$-simplex that converges in probability to a dense subset of the $(\#(\mathcal{U}^{(0)} \times K^0) - 1)$-simplex as $T \to \infty$.

The limit theory of this testing procedure is derived in the following result.

**Theorem 4.** *Suppose that Assumption 1 and conditions (c) and (d) of Theorem 3 hold, $\epsilon \to 0$ with T, while, $\mathcal{U}_\epsilon^{(0)}$ converges in the Painleve–Kuratowski topology to a non-stochastic, dense subset of $\mathcal{U}^{(0)}$ in probability, $S_T$ converges in probability in the Painleve–Kuratowski topology to a dense, non-stochastic subset of the $(\#(\mathcal{U}_\infty^{(0)} \times K^0) - 1)$-simplex and the spectrum of $\mathrm{Var}\left[ \sum_{u \in \mathcal{U}_\epsilon^{(0)}} \sum_{\kappa \in \mathrm{K}^{(0)}} \beta_{u,\kappa} \left( u\left(x_0^{\mathrm{T}}(\kappa + \delta)\right) - u\left(x_0^{\mathrm{T}}\kappa\right) \right) \right]_{\boldsymbol{\beta}}$, where the parameter $\boldsymbol{\beta}$ lies in the set $\left\{ \boldsymbol{\beta} : \sum_{u \in \mathcal{U}_\infty^{(0)}} \sum_{\kappa \in \mathrm{K}^{(0)}} \beta_{u,\kappa} D(u_j, \kappa_i, \delta, \mathbb{P}) = 0 \right\}$, is bounded away from zero; here, $\mathcal{U}_\infty^{(0)}$ represents the Painleve–Kuratowski limit in probability of $\mathcal{U}_\epsilon^{(0)}$. Then, the following results are obtained:*

*(i) under* $\mathbf{H}_0$,

$$\limsup_{T \to \infty} \mathbb{P}\left( (\mathrm{ELR}_T^\star - N_1(\delta, \mathbb{P}_T, c_T)) / \sqrt{2N_1(\delta, \mathbb{P}_T, c_T)} > \left( q(1-\alpha, \chi^2_{N_1(\delta, \mathbb{P}_T, c_T)}) - N_1(\delta, \mathbb{P}_T, c_T) \right) / \sqrt{2N_1(\delta, \mathbb{P}_T, c_T)} \right) \le \alpha. \tag{16}$$

*(ii) under* $\mathbf{H}_1$, *and if, moreover,* $\mathrm{card}\left( \mathcal{U}_\epsilon^{(0)} \right) = o_p(\frac{T}{B})$,

$$\lim_{T \to \infty} \mathbb{P}\left( (\mathrm{ELR}_T^\star - N_1(\delta, \mathbb{P}_T, c_T)) / \sqrt{2N_1(\delta, \mathbb{P}_T, c_T)} > \left( q(1-\alpha, \chi^2_{N_1(\delta, \mathbb{P}_T, c_T)}) - N_1(\delta, \mathbb{P}_T, c_T) \right) / \sqrt{2N_1(\delta, \mathbb{P}_T, c_T)} \right) = 1. \tag{17}$$

**Proof of Theorem 4.** Analogous to the proof of Theorem 3, the distance between the number of contacts in $S_T$ and in its PK-limit can be shown to converge to zero in probability. Using this and arguments analogous to the ones concerning empirical process convergence, Skorokhod representations and bounds on infima of quadratic forms over cones in the proof of Theorem 4.2.1 of [30], under the null, it can then be shown that the test statistic is asymptotically bounded from above by a random variable that weakly converges to a standard normal. Under the alternative, the relevant part of the proof of Theorem 4.2.1 of [30] implies that $\mathrm{ELR}_T^\star \ge O_p(\frac{T}{B})$. Thereby, the growth condition on the approximation of $\mathcal{U}$ along with (c) imply that the test statistic diverges to infinity. $\square$

The resulting test is also asymptotically conservative and consistent. It extends the ELR test of [18] as it allows for non-singleton K and temporal dependence for the underlying stochastic processes. Whenever $\mathcal{U} \ne \mathrm{cl}(\mathcal{U})$, the bounded away from zero spectrum condition is similar to the restricted stochastic dominance analysis of [18]; $\mathcal{U}$ may need reduction when its boundary contains utilities associated with trivial contacts, to ensure existence of a well-defined limiting distribution. Whenever the set of Russell and Seo utilities is used-see [21], this can be performed similarly to the analysis described above for the modification of the BELR test for the null and for the case of infinite contacts.

### 4. Monte Carlo Simulations

A simulation study is performed to obtain an impression of the test properties in finite samples in a controlled environment. The focus is on the effects of the cross-sectional dimension ($N$) and time-series dimensions ($T$), which are likely to play a key role in the research design of most empirical studies. We therefore use a basic specification with $\mathcal{U} = \mathcal{U}_2$ (all globally risk-averse investors) and a tractable, bounded and serially iid probability distribution (without the need for the blockwise approach). A follow-up study could also simulate the effects of the set of preferences, the shape of the distribution and the data dynamics.

The arbitrage set $\Delta$ consists of one element ($\delta$) with a uniform payoff distribution: $x^{\mathrm{T}}\delta \sim U[-0.5 - c, 0.5 - c]$, $c \in \{0, 0.05\}$. The set of hosts K consists of all convex mixtures of $N = 1, 2, 4, 8$ extreme elements with a joint distribution such that the combined portfolios are mutually iid: $x^{\mathrm{T}}\kappa_i + x^{\mathrm{T}}\delta \sim U[0, 1]$, $i = 1, \cdots, N$.

For $c = 0$, the overlay qualifies as an SAO because it is a mean-preserving anti-spread for all hosts. Non-trivial contact points then occur at the support upper bound $x = 1.5$; the number of non-trivial population contacts is thus equal to $N$. Trivial contacts also occur at the support's infimum $x = -0.5$, but nowhere else.

For $c = 0.05$, the overlay does not qualify because it reduces the means of the hosts. The value $c = 0.05$ is relatively small compared with the mean and dispersion of the hosts, so the distribution lies not too deep in the alternative hypothesis. Non-trivial contact points now occur at a unique point in the interior of the support; the number of non-trivial contacts is again $N$.

The distribution obeys the assumption framework for applying our ELR test to individual observations instead of data blocks ($B = L = 1$). For applying the test, the support of the hosts ($[-0.5 + c, 1.5 - c]$) is discretized using a regular grid with 30 grid points. For estimating the number of contact points, using Theorem 3 and some experimentation, we define $c_T = 10^{-11}/(\ln(T)\ln(N))$. The minimum number is set at one, usingf the insight that optimized overlays have at least one binding stochastic enhancement constraint. Furthermore, to avoid trivial contacts that would invalidate the minimum eigenvalue condition of Theorem 3, we truncate the left tail of the host distribution at $-0.4 + c$.

We draw 1000 random samples of size $T = 10^1, 10^2, 10^3, 10^4$, and apply the ELR test for the null hypothesis that the overlay is qualified, using a nominal significance level of 0.05, for $N = 1, 2, 4, 8$. The rejection rate for $c = 0$ measures the test size (relative frequency of false rejections over the 1000 samples); the rejection rate for $c = 0.05$ measures the test power (relative frequency of correct rejections over the 1000 samples).

Information about the quality of the moment selection procedure is provided in the form of the average (over the 1000 samples) of the number of estimated non-trivial contacts. If the estimated number is larger than the population number ($N$), then the procedure compromises test power.

Table 1 summarizes the simulation results. The conservative nature of the test is clearly visible: for all data dimensions ($T, N$), the test size is under control. The test size increases with the sample size, and it mostly decreases with $N$. It flattens in large samples at levels well below the nominal significance level of 0.05. The test power becomes acceptable to good for a sample size between $T = 300$ and $T = 500$, and it is very good for $T \geq 1000$.

The average estimated number of contact points suggests that the test performance can be further improved by refining the moment selection procedure. For $T = 10$, the number of population contacts is overestimated by a factor of almost five, which partly explains the low test power for this sample size. In contrast, for $T = 10^4$, the estimates are approximately one-half of their population counterparts. The results suggest that the test performance can be improved by reducing the rate at which the slacks converge to zero with $T$.

**Table 1.** Monte Carlo results.

| | Monte Carlo Size ($c = 0$) | | | | Monte Carlo Power ($c = 0.05$) | | | |
|---|---|---|---|---|---|---|---|---|
| | $N = 1$ | $N = 2$ | $N = 4$ | $N = 8$ | $N = 1$ | $N = 2$ | $N = 4$ | $N = 8$ |
| $T = 10$ | 0.003 | 0.000 | 0.001 | 0.000 | 0.011 | 0.001 | 0.002 | 0.000 |
| $T = 10^2$ | 0.006 | 0.000 | 0.000 | 0.000 | 0.200 | 0.063 | 0.005 | 0.000 |
| $T = 10^3$ | 0.029 | 0.007 | 0.000 | 0.000 | 1.000 | 0.998 | 0.994 | 0.947 |
| $T = 10^4$ | 0.023 | 0.025 | 0.004 | 0.000 | 1.000 | 1.000 | 1.000 | 1.000 |

| | Average Number of Selected Moments | | | | | | | |
|---|---|---|---|---|---|---|---|---|
| | $c = 0$ | | | | $c = 0.05$ | | | |
| | $N = 1$ | $N = 2$ | $N = 4$ | $N = 8$ | $N = 1$ | $N = 2$ | $N = 4$ | $N = 8$ |
| $T = 10$ | 5.8 | 11.132 | 23.031 | 46.374 | 5.952 | 11.433 | 23.679 | 47.543 |
| $T = 10^2$ | 2.176 | 4.363 | 8.655 | 17.274 | 2.257 | 4.523 | 9.005 | 18.006 |
| $T = 10^3$ | 1.044 | 2.011 | 3.964 | 7.914 | 1.055 | 2.036 | 4.007 | 8.016 |
| $T = 10^4$ | 1 | 1.376 | 2.461 | 4.866 | 1 | 1.376 | 2.461 | 4.866 |

Entries in the upper panel report the null hypothesis rejection rate over the Monte Carlo replications for $c = 0$ (Monte Carlo size) and the analogous rejection rate for $c = 0.05$ (Monte Carlo power), for every pair $(T, N)$ where $T = 10^1, 10^2, 10^3, 10^4$, $N = 1, 2, 4, 8$. Entries in the lower panel report the average over the Monte Carlo replications of the estimated number of non-trivial contacts for those tests, where the slacks used are $c_T = 10^{-11}/(\ln(T) \ln(N))$.

## 5. An Empirical Exercise

Ref. [13] empirically test whether each of several overlays formed in the US stock market is an SAO for all globally risk-averse investors ($\mathcal{U} = \mathcal{U}_2$) who hold an equity portfolio from a host set ($K$). The existence of SAOs would represent empirical evidence against stock market efficiency that is robust to the specification of the endowments of the investors; it would also provide a motivation for adopting active investment strategies that resemble the overlays that are classified as SAOs.

The evaluated overlays are seven equity factor portfolios from the empirical finance literature (SMB, HML, RMW, CMA, STR, ITM and LTR) and several mixtures of the factor portfolios, namely the equally weighted portfolio (EWP), a portfolio that is constructed using optimization subject to empirical mean-variance constraints (MVP), and a portfolio that is optimized using empirical stochastic enhancement constraints (SDP).

For brevity, we only report here the analysis of the most challenging case in which the host set consists of all convex mixtures of 10 standard industry portfolios represented by the nine-dimensional standard simplex: $K = K_{10} = \{\kappa \in \mathbb{R}^{10} : \kappa^T \mathbf{1}_{10} = 1; \kappa \geq \mathbf{0}_{10}\}$. The original study also covered the simpler case with a single host ($K = K_1$) equal to the value-weighted stock market portfolio. This case seems less interesting here because it amounts to a simple pairwise comparison between the market portfolio and the combination of the market portfolio and the evaluated overlay.

The out-of-sample analysis is based on monthly percentage total returns for three non-overlapping subperiods of 15 years, or $T = 180$ months: 1978–1992, 1993–2007 and 2008–2022. The observations show significant violations of iidness in the form of volatility clustering. For this reason, the ELR test is implemented in a blockwise manner. The appropriate blocking structure is not obvious because we don't have accurate estimates of the volatility-of-volatility and the mean reversion rate of volatility. The original study is extended here by implementing the BELR tests using various alternative specifications of $(B, L)$, with $B = 1, 3, 6, 12$ and $1 \leq L \leq B$.

The results in Table 2 show an important reduction in many *p*-values for large blocks ($B = 12$) with limited overlap ($L = 12, 9, 6$), especially in the second sample period (1993–2017). It is not clear whether the higher rejection rates reflect the reduced number of blocks or the data dependence due to the limited length of the available time series. None of the evaluated overlays, including the one based on optimization with empirical stochastic enhancement constraints, can therefore be classified as an SAO in every sample period and for every blocking structure.

The reported sensitivity of the *p*-values underlines the importance of selecting a proper blocking structure and a proper moment estimation method for constructing the rejection region and *p*-values in small samples. The results also suggest that the optimization approach to building SAOs may benefit from tightening the empirical enhancement constraints. Naturally, a higher confidence in the stochastic enhancement constraints generally has to be balanced against the possible opportunity loss from adopting suboptimal solutions.

**Table 2.** Blockwise ELR test results.

|      |      | B | L | SMB | HML | RMW | CMA | STR | ITM | LTR | EWP | MVP | SDP |
|------|------|---|---|-----|-----|-----|-----|-----|-----|-----|-----|-----|-----|
| 1978 | 1992 | 1 | 1 | 0 | 0.964 | 0 | 0.985 | 0 | 0 | 0 | 0 | 0 | 0.969 |
|      |      | 3 | 3 | 0 | 0.940 | 0 | 0.974 | 0 | 0 | 0 | 0 | 0 | 0.947 |
|      |      | 6 | 6 | 0 | 0.917 | 0 | 0.964 | 0 | 0 | 0 | 0 | 0 | 0.924 |
|      |      | 6 | 3 | 0 | 0.905 | 0 | 0.962 | 0 | 0 | 0 | 0 | 0 | 0.922 |
|      |      | 12 | 12 | 0 | 0.837 | 0 | 0.948 | 0 | 0 | 0 | 0 | 0 | 0.893 |
|      |      | 12 | 9 | 0 | 0.602 | 0 | 0.651 | 0 | 0 | 0 | 0 | 0 | 0.607 |
|      |      | 12 | 6 | 0 | 0.861 | 0 | 0.942 | 0 | 0 | 0 | 0 | 0 | 0.880 |
|      |      | 12 | 3 | 0 | 0.863 | 0 | 0.947 | 0 | 0 | 0 | 0 | 0 | 0.890 |
| 1993 | 2007 | 1 | 1 | 0 | 0.579 | 0.690 | 0.587 | 0 | 0.573 | 0.692 | 0.869 | 0.800 | 0.637 |
|      |      | 3 | 3 | 0 | 0.224 | 0.502 | 0.402 | 0 | 0.348 | 0.568 | 0.771 | 0.662 | 0.254 |
|      |      | 6 | 6 | 0 | 0.122 | 0.366 | 0.260 | 0 | 0.114 | 0.419 | 0.659 | 0.535 | 0.151 |
|      |      | 6 | 3 | 0 | 0.106 | 0.356 | 0.229 | 0 | 0.184 | 0.404 | 0.663 | 0.529 | 0.134 |
|      |      | 12 | 12 | 0 | 0.031 | 0.206 | 0.088 | 0 | 0 | 0.227 | 0.504 | 0.384 | 0.027 |
|      |      | 12 | 9 | 0 | 0.000 | 0.060 | 0 | 0 | 0 | 0.441 | 0.256 | 0.067 | 0.001 |
|      |      | 12 | 6 | 0 | 0.020 | 0.157 | 0.077 | 0 | 0 | 0.189 | 0.497 | 0.328 | 0.029 |
|      |      | 12 | 3 | 0 | 0.036 | 0.207 | 0.100 | 0 | 0.024 | 0.235 | 0.516 | 0.362 | 0.047 |
| 2008 | 2022 | 1 | 1 | 0 | 0 | 0.817 | 0.835 | 0 | 0.679 | 0 | 0 | 0.897 | 0.682 |
|      |      | 3 | 3 | 0 | 0 | 0.695 | 0.705 | 0 | 0.463 | 0 | 0 | 0.827 | 0.512 |
|      |      | 6 | 6 | 0 | 0 | 0.565 | 0.608 | 0 | 0.282 | 0 | 0 | 0.767 | 0.378 |
|      |      | 6 | 3 | 0 | 0 | 0.558 | 0.619 | 0 | 0.307 | 0 | 0 | 0.730 | 0.420 |
|      |      | 12 | 12 | 0 | 0 | 0.451 | 0.539 | 0 | 0 | 0 | 0 | 0.681 | 0.222 |
|      |      | 12 | 9 | 0 | 0 | 0.556 | 0.467 | 0 | 0 | 0 | 0 | 0.722 | 0.383 |
|      |      | 12 | 6 | 0 | 0 | 0.631 | 0.516 | 0 | 0 | 0 | 0 | 0.693 | 0.408 |
|      |      | 12 | 3 | 0 | 0 | 0.420 | 0.505 | 0 | 0.118 | 0 | 0 | 0.634 | 0.258 |

Entries report the *p*-values for the BELR tests for the hypotheses of each of the seven equity factor portfolios (SMB, HML, RMW, CMA, STR, ITM, LTR) and three mixtures of the factor portfolios (EWP, MVP, SDP). Here, the host set is the standard nine-dimensional simplex of all mixtures of ten standard industry portfolios ($K = K_1 0$). The tests are performed across three sub-samples (1978–1992, 1993–2007, 2008–2022) and for different choices of block parameters (*B* and *L*).

## 6. Discussion

Generalizing the work of [9], we obtained versions of set consistency for empirical SAOs. The results provide a theoretical basis for empirical tests for the existence of SAOs and the use of portfolio optimization under empirical SAO restrictions in large samples.

The analysis avoided the use of (arbitrary) tolerance parameters in the empirical optimization problem to obtain consistency. This feature is beneficial because the proper specification of those nuisance parameters is application-specific, empirical results are sometimes not robust to the specification, and the calibration to specific applications increases the computational burden. To avoid the use of tolerances, our analysis used a set of approximate solutions that satisfy stricter versions of the empirical dominance inequalities.

We designed a procedure for statistical inference about membership/non-membership of the SAO set via BELR tests. Two tests were developed: one for the null that a given overlay is an SAO, and another for the alternative that the portfolio is not an SAO.

Special attention was given to the estimation of the contact sets that define the number of degrees of freedom for the chi-squared distributions for the two tests. We also introduced robust versions of the testing procedures that allow for the possibility that the set of binding dominance inequalities is infinite.

For the BELR tests, we have derived asymptotic conservatism and consistency. Naturally, these asymptotic results do not suffice to establish favorable test properties in applications with realistic data dimensions, especially if the set of utility functions $\mathcal{U}$ is large. Simulations and applications can shed light on this important topic.

The results of the simulation study pointed to favorable statistical size and power properties for testing the null, for realistic data dimensions $(N, T)$. These results are particularly promising because they are based on the most general specification of the risk preferences $(\mathcal{U} = \mathcal{U}_2)$; the test is expected to be more powerful for higher-degree SD criteria $(\mathcal{U} = \mathcal{U}_n)$, $n > 2$.

It seems interesting to perform a follow-up study that is devoted to additional simulations. The follow-up could calibrate the shape of the marginal distributions and the mutual correlation structure to several representative empirical applications. It could also analyze the effects of the utility function class $(\mathcal{U})$, the data dynamics and blocking structure, and the moment estimation method.

Due to the conservative approach, the test may perform poorly on elements of the alternative hypothesis that are close to the boundary between the two hypotheses. This problem can be partially addressed by combining the test for the null with the test for the alternative. To conclude that an overlay is an SAO, it seems useful to check whether the first test is passed (no rejection of being an SAO) and the second test is not passed (rejection of not being an SAO). If both tests are passed or both tests are not passed, then the data set neither confirms nor challenges the prior beliefs that the analyst holds about the overlay.

Refinements of the BEL test strategies with better power under local alternatives are on our research agenda. One such approach could combine (i) analytical approximations of the right tails of the limiting null distributions based on topological invariants like the Euler characteristic (see [40]) and (ii) generalized moment selection procedures (see [35]). Attractively, this approach would preserve the computational efficiency of using conservative chi-squared rejection regions compared with resampling methods.

Follow-up research could also consider the selection of the appropriate blocking scheme and moment estimation method, given the sensitivity of the results and conclusions in the empirical analysis of factor portfolios and the lack of accurate estimates of the data dynamics.

The focus has been on the empirical counterpart of the unconditional distribution. Extensions are possible to situations in which $\mathbb{P}$ is conditioned on economic or financial variables or $\mathbb{P}$ is estimated using other non-parametric methods or (semi-)parametric models that are more statistically efficient (at the possible risk of introducing specification error).

The estimation could rely on a (semi-)parametric model or a fully non-parametric method, depending on the data dimensions and features and on possible prior information. An example of (semi-)parametric estimation is (quasi) maximum likelihood; an example of an alternative non-parametric estimation is Nadaraya–Watson. To obtain statistical consistency in these cases, it is sufficient to assume that the relevant likelihood function is smooth or to assume standard regularity properties for the unknown densities involved and the associated kernels. Parametric likelihood ratio tests and smoothed empirical likelihood tests can replace the BELR test in these cases.

Although our initial motivation was to handle multiple hosts, the analysis applies a fortiori to the common special case in which K is a singleton and the ELR test becomes a test for pairwise dominance. Although the test is conservative, it may prove to have advantages compared to existing tests for pairwise dominance based on statistical resampling methods, e.g., the subsampling tests in [28]. For example, if the data are iid, the results of [41] (see Theorem 3.2) and of [42] (see Theorem 7.1.3) seem to imply large deviation advantages of our ELR test—at least for elements of the alternative that are sufficiently "far" from the null of being an SAO—compared to the existing tests.

Extending the comparison between our conservative BELR tests and resampling methods to non-iid settings is of interest given the data dependence in typical investment applications and the flexibility of the proposed BEL apparatus. In future work, we therefore hope to establish a large-deviation property for blocks of stationary mixing data with a rate function (see, for example, [42]) that directly depends on the KL divergence between distributions on a space of limiting blocks. The form of the rate function (if it exists) could

be connected to some sort of large-deviation optimality of the conservative BEL approach for appropriate elements of the alternative hypothesis.

The test for the alternative avoids pathologies of asymptotic non-tightness for the limiting alternative distribution by focusing on an alternative system of inequalities that consider convex combinations of the original stochastic dominance inequalities. It extends the ELR test for pairwise non-dominance of [18], since it allows for non-uniqueness of the host portfolio and stationarity and mixing for the stochastic process.

The hope is that our consistency results and statistical tests will contribute to the further proliferation of the SD concept in asset pricing, portfolio management and other possible application areas.

The use of multiple hosts seems particularly useful for analyzing hedge funds and financial derivatives because there is generally no natural benchmark or host in these cases. For example, the Chicago Board of Options Exchange forwards a myriad of Strategy Benchmark Indices for stock index options.

The analysis also applies to alternative application areas where sets of risky alternatives are compared based on time-series estimates of the joint distribution functions. One such application area is the evaluation and combination of forecast models in forecasting; see, e.g., [30]. Encouragingly, the use of non-monotonic loss functions (outside $\mathcal{U}_2$) in forecasting does not affect the validity of our derivations and arguments.

Another interesting route for further research is to adjust the analysis to cases where the underlying probability distribution is estimated using cross-sectional data, such as the comparison of empirical well-being distributions in well-being analyses, in the spirit of [43].

**Author Contributions:** Conceptualization, S.A. and T.P.; methodology, S.A. and T.P.; software, S.A. and T.P.; validation, S.A. and T.P.; formal analysis, S.A. and T.P.; investigation, S.A. and T.P.; data curation, S.A. and T.P.; writing—original draft preparation, S.A. and T.P.; writing—review and editing, S.A. and T.P.; supervision, S.A. and T.P.; project administration, S.A. and T.P. All authors have read and agreed to the published version of the manuscript.

**Funding:** This research received no external funding.

**Data Availability Statement:** The authors confirm that the data and materials that support the results or analyses presented in this paper are freely available upon request.

**Conflicts of Interest:** The authors declare no conflicts of interest.

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
