# Peer review of "Stochastic Arbitrage Opportunities: Set Estimation and Statistical Testing"

_mathematics, doi:10.3390/math12040608_

Round 1
Reviewer 1 Report
Comments and Suggestions for Authors
File the attached

Conduct a thorough proofreading to catch minor grammatical errors, typos, or punctuation issues. This will further elevate the professionalism of the manuscript.
Author Response
Please look at the second part of the uploaded pdf file.

Reviewer 2 Report
Comments and Suggestions for Authors
This paper presents a formal theory of consistent estimation of the set of all arbitrage portfolios that meet the description of being a Stochastic Arbitrage Opportunity. Two Empirical Likelihood Ratio tests are developed: one for the null that a given arbitrage portfolio is qualified and another one for the alternative that the portfolio is not qualified. While the contents of the paper demonstrate sufficiency and present novel results, there are areas for improvement.
1. It would be better to Include some practical examples or case studies to illustrate the application of the statistical theory and optimization algorithms would be beneficial. Real-world examples would help readers understand how these concepts are applied in empirical asset pricing and delegated money management.
2. To maintain the paper's relevance, it is imperative to include discussions on recent advances in parametric estimation methods. Notably, insights from Wang et al.'s regression analysis of clustered panel count data (Statistical Papers, 2023, DO1:10.1007/s00362-023-01511-3) and Luo et al's work on system reliability modeling (Reliability Engineering and System Safety, 2022, 218.108136) should be integrated. This inclusion will underscore the paper's connection to contemporary research.
3. Some simulation studies should be provided to assess the finite-sample properties of the proposed estimators and tests.
Author Response
Please look at the third part of the uploaded pdf.

Round 2
Reviewer 2 Report
Comments and Suggestions for Authors
The authors have revised the paper well.